# Modeling Approaches Reveal New Regulatory Networks in *Aspergillus fumigatus* Metabolism

**DOI:** 10.3390/jof6030108

**Published:** 2020-07-14

**Authors:** Enzo Acerbi, Marcela Hortova-Kohoutkova, Tsokyi Choera, Nancy Keller, Jan Fric, Fabio Stella, Luigina Romani, Teresa Zelante

**Affiliations:** 1Nlytics Pte. Ltd., Singapore 637551, Singapore; contact@nlytics.ai; 2Centre for Translational Medicine, International Clinical Research Centre, St. Anne’s University Hospital Brno, 65691 Brno, Czech Republic; marcela.hortova@fnusa.cz (M.H.-K.); jan.fric@fnusa.cz (J.F.); 3Department of Medical Microbiology and Immunology, Department of Bacteriology, University of Wisconsin, Madison, WI 53706, USA; tchoera@hexagonbio.com (T.C.); npkeller@wisc.edu (N.K.); 4Institute of Hematology and Blood Transfusion, 12800 Prague, Czech Republic; 5Department of Informatics, Systems and Communication, University of Milano-Bicocca, Viale Sarca 336, Building U14, 20126 Milan, Italy; fabio.stella@unimib.it; 6Department of Experimental Medicine, University of Perugia, 06132 Perugia, Italy; luigina.romani@unipg.it

**Keywords:** *Aspergillus fumigatus*, tryptophan metabolism, modeling, Bayesian networks, continuous time Bayesian networks, gene network reconstruction, gene network inference

## Abstract

Systems biology approaches are extensively used to model and reverse-engineer gene regulatory networks from experimental data. Indoleamine 2,3-dioxygenases (IDOs)—belonging in the heme dioxygenase family—degrade l-tryptophan to kynurenines. These enzymes are also responsible for the de novo synthesis of nicotinamide adenine dinucleotide (NAD+). As such, they are expressed by a variety of species, including fungi. Interestingly, *Aspergillus* may degrade l-tryptophan not only via IDO but also via alternative pathways. Deciphering the molecular interactions regulating tryptophan metabolism is particularly critical for novel drug target discovery designed to control pathogen determinants in invasive infections. Using continuous time Bayesian networks over a time-course gene expression dataset, we inferred the global regulatory network controlling l-tryptophan metabolism. The method unravels a possible novel approach to target fungal virulence factors during infection. Furthermore, this study represents the first application of continuous-time Bayesian networks as a gene network reconstruction method in *Aspergillus* metabolism. The experiment showed that the applied computational approach may improve the understanding of metabolic networks over traditional pathways.

## 1. Introduction

Microbial metabolism is under deep investigation because of the recent advances in enabling metagenomic technologies and the urgent need to further understand the functions of microbes able to colonize human tissues. At this stage, information related to different metabolic pathways and advances in metabolomics represent an important tool for drug development and target discovery. Mathematical modeling has been recently applied to predict potential interactions between different pathways in the yeast *Saccharomyces cerevisiae* [1]. In particular, the production of immunomodulatory metabolites by the fungus *Aspergillus fumigatus* is of great interest for the impact on the host immune system [2,3,4].

In this contest, the biological functions of several xenobiotic receptors, which are able to recognize microbial metabolites, have been investigated [5]. The Aryl hydrocarbon receptor (AhR), for example, is able to recognize several products of the aminoacidic catabolism, as well as mycotoxins [5,6].

Of interest, in mammalians the activation of AhR in the gut has been proved to monitor anti-inflammatory responses, with protective effects as in candidiasis or in inflammatory bowel disease. In other tissues, such as the pulmonary tract, the role of xenobiotic receptors is very complex and still under debate [7].

The fungus *A. fumigatus* is able to degrade/utilize the essential amino acid Tryptophan (Trp) into distinct metabolites following three main pathways. One is via the known rate limiting enzyme aromatic aminotransferase (Aro), one via indoleamine 2,3-dioxygenase (Ido) activity, and a third not associated with a specific enzyme but rather several secondary metabolite enzymes that place Trp and prenylated Trp into small non-ribosomally encoded secondary metabolites. Among the *aro* genes, *aroH* (Afu2g13630) encodes the putative pyridoxal 5′-phosphate (PLP)-dependent aromatic aminotransferase, which transforms Trp into indolepyruvate [4,8]. *Ido* genes (*idoA* Afu3g14250, *idoB* Afu4g09830, *idoC* Afu7g02010) encode putative IDOs, which transform Trp to l-kynurenine. The secondary metabolites pathways yield four known toxins, fumiquinazoline, fumitremorgin, fumigaclavine and hexadehydroastechrome [9]. Although the steps of these different metabolite pathways are well-known, the interactions between the two catabolic axes (Ido and Aro) remain unclear. In this study, the generation of high granularity time-course data allowed for a computational analysis of the dynamics of interactions among the genes in these two systems.

The task of uncovering the causal structure (under the following assumptions: causal sufficiency, faithfulness and the causal Markov condition) of regulatory interactions (often referred to as “gene regulatory networks”or GRNs) is a fervent area of research in computational biology [10,11]. A number of approaches have been applied to the GRNs reconstruction problem. Probabilistic graphical models such as Bayesian networks [12] were shown to be powerful tools for solving the GRN reconstruction problem [13], and they led to significant discoveries [14]. When richer time-course measurements started to be made available, Dynamic Bayesian networks (DBNs) gained more and more relevance in the field. Other probabilistic approaches are state space models [15] and probabilistic Boolean networks; [16] however, it has been shown that the latter are outperformed by DBNs for GRN reconstruction problems [17]. Granger causality (GC) is a robust method for analyzing time-course data; since its early introduction, it has been successfully applied to a multitude of domains, such as economics, neuroscience and biology. Continuous-time Bayesian networks (CTBNs) are an emerging approach, which, thanks to their explicit representation of the time, provide state-of-the-art performance for the problem of gene network reconstruction when time-course data are available [18]. Weighted gene co-expression network analysis (WGCNA) is another widely applied methodology. Unlike WGCNA, which is based on pairwise correlation relationships among genes, CTBNs are based on detecting relationships of causality [19] among random variables whose state evolves over time. CTBNs were proven to be a comparable choice to both GC and DBNs for small-scale networks and a preferable choice to both GC and DBNs for networks of large size and when measurements are collected at unevenly spaced time points [18]. A recent review of existing network reconstruction approaches can be found in [20].

Nowadays, the finely grained time-course data generated by high throughput technologies are particularly suitable for computational methodologies which are conceived to exploit the dynamic nature of datasets, like CTBNs. CTBNs have been recently applied to the analysis of molecular data, to investigate the regulatory interactions that characterize pathogenic versus non-pathogenic murine TH17 cells [18] and TH17 cell differentiation in humans, where their application led to the discovery of a new regulator gene [21]. The graphical component of a CTBN provides an intuitive level of abstraction (in the form of a network) of how the regulatory process operates over the duration of the experiment: nodes corresponds to genes, and arcs represent direct probabilistic relationships among genes (one gene exerting a direct influence over the other). In this study, the structure of regulatory interactions controlling Trp metabolism was inferred by using CTBNs.

## 2. Materials and Methods

**Strains and medium.** The strains that were used in this study are listed in Table 1. The genetic background of the primary strain used in this study was *A. fumigatus* CEA17 (Table 1). All strains were maintained as glycerol (Panreac, Miami, Florida, USA) stocks, at −80 °C, and activated on solid glucose minimal media (GMM), at 37 °C [22].

**Genetic manipulations for *A. fumigatus aroH* mutants by protoplasting method.** Fungal DNA extraction, gel electrophoresis, restriction enzyme digestion, Southern blotting, hybridization and probe preparation were performed according to standard methods [24]. For DNA isolation, *A. fumigatus* strains were grown for 24 h at 37 °C, in static liquid GMM. DNA isolation was performed as described by Sambrook and Russell [24]. Gene deletion mutants in this study were constructed by targeted integration of the deletion cassette through transformation [25,26]. The deletion cassettes were constructed by using a double-joint fusion PCR (DJ-PCR) approach [25,26]. *A. fumigatus* protoplast generation and transformation were carried out as previously described [25,26].

**Fungal cell culture.** Fungal strains were put in culture (1 × 10^8^ conidia/mL) in RPMI 1640 medium (GIBCO, Milano, Italy), for each condition, at 37 °C. Supplemental Trp (Sigma-Aldrich Merck Life Science S.r.l. Milano, Italy) resulted in a final concentration of 100 μM. Cells were harvested for RNA isolation every 10 min for 3 h.

**RNA isolation and qPCR.** Fungal biomass was disrupted by using a FastPrep-FP120 (BIO101) (Qbiogene, Inc, Illkirch, CEDEX, France) at 4.5 m/s for 30 s. Samples were left at room temperature for 5 min, in ice, and subsequently centrifuged for 10 min at 13,000 rpm at 4 °C. Total RNA was extracted from purified cells by the TRIzol method (Invitrogen, Milano, Italy), according to the manufacturer’s protocol. The cDNA was synthesized by using the PrimeScript RTreagent kit (TAKARA, Saint-Germain-en-Laye France). Then, qPCR was carried out with primers listed in Table 2, using SybrGreen Expression Master Mix (Thermo Fisher Scientific, Milano, Italy). The qPCR analysis was performed by using a LightCycler II (Roche, Basel, Switzerland). The Ct values of genes of interest were normalized to house-keeping gene 18S (ΔCt), and the relative expression of each gene of interest was calculated as 2^−ΔCt^. All reactions were repeated at least three times, independently, and normalized with *β-actin* gene expression.

**Data preprocessing and learning parameters.** Raw data were analyzed by using R version 3.1.2 and the Bioconductor package. Data were log2 transformed. Four separate networks were learned by using time-course datasets from the experiments wild type, wild type with the addition of Trp, wild type with *aroH* mutant and wild type with *aroH* knockout with the addition of tryptophan. Missing values in time-course datasets were replaced with zeroes. The R package ctbn v.1.0 was used for the analysis. Due to the limited amount of data available for each experimental condition, structural learning of continuous-time Bayesian networks (CTBNs) was run multiple times, with varying hyperparameter combinations. Specifically, *α* was tested for values equal to 3, 2, 1, 0.1, 0.01 and 0.001, while *τ* for values equal to 10, 5, 1, 0.1 and 0.01. The experimental campaign included testing various discretization approaches, eventually choosing to discretize the data into 3 bins of equal size. This resulted in 30 candidate networks learned for each of the 4 experimental conditions. Only arcs detected in more than 90% of candidate networks that had at least one arc were considered to be high-confidence and reported in the final graphs shown in Figure 3.

**Statistical analysis.** Statistical analysis was performed with ANOVA tests and GraphPad Prism 6 software (GraphPad Software, San Diego, CA, USA).

## 3. Results and Discussion

The network shown in Figure 1 represents the known, literature-based Trp catabolic cascade in *A. fumigatus,* which is organized in two separate axes and activated by Trp. One axis is regulated by the *ido* genes, and a secondary axis is regulated by the *aroH* gene, in a way that each gene controls the transcription of another gene downstream in the same axis.

The axis regulated by the *ido* genes leads to kynurenine production. Those metabolites are very well-known in the mammalian metabolism as important modulators of the “Trp starvation” response in inflammation and infection. In particular, l-Kynurenine is known to bind the xenobiotic receptor AhR and reduce T-cell reactivity [27,28]. In addition, when Trp starvation is induced by IDO expression, effector T cells undergo anergy, reducing inflammation. In infections, Trp starvation is also described to reduce microbial metabolism and pathogen virulence [29]. In the fungus *Aspergillus*, the role of Trp metabolism is still under investigation, although several studies already characterized the protein functions of the catabolic cascade [30].

Idos proteins in *Aspergillus* seem to degrade Trp in to kynurenines, with different affinity for Trp and velocity rate [8]. The Aro pathway in *Aspergillus* is less characterized, although we have recently characterized the protein function, and we have shown that the catabolic cascade induces the release of indole derivatives as indole acetate from Trp (manuscript submitted) [31].

However, all of these studies highlight the fundamental concept that Trp metabolism can be activated off or on by culturing cells in low-Trp or in high-Trp concentration, respectively.

Therefore, we placed in culture the fungus *Aspergillus* in conditions of low or high Trp availability, in order to switch off or strongly activate the whole cascade, respectively. The scheme of the experiment and the forward analysis based on expression data are shown in Figure 2. Knowing that both pathways may potentially degrade Trp [4,31], we maximally induced the catabolism on the other axis by removing one axis, as shown by qPCR results (Appendix A
Appendix A).

For this purpose, we used the comparison between two different strains of fungi: the wild-type CEA17 and the mutant strain of *Aspergillus* for the gene *aroH* (see Table 1), obtained by genetic manipulation, using the protoplasting method. The obtained mutants were named in this study TTC22 × (Δ*aroH*).

We selected the removal of the well-characterized rate limiting enzyme Aro for one axis (“condition B”) (Figure 2) in comparison with “condition A”, where, in the wild-type strain, both axes are active and able to degrade Trp. As mentioned above, for “condition A”, we used the wild-type strain CEA17 *pyrG*- KU80, and for “condition B”, we used the mutant strain ∆*aroH*.

In order to analyze the dynamic behavior of the network, we performed experiments by placing cells on low Trp (off experiments) to high Trp (on experiments). We collected samples every 10 min, for up to 3 h. We analyzed expression profiles of catabolic genes by quantitative real-time RT-PCR (q-PCR), focusing on four genes (*idoA*, *idoB*, *idoC*, and *aroH*). Thus, our network was indeed composed of four different genes, which codify for catabolic enzymes of the Trp cascade.

In the off experiments, Trp availability slightly led to the transcription of all the network genes by inducing two waves of expression for *idos* and only one wave of expression for *aroH.* Their dynamic behavior is obvious, as in the *on* experiments, a seemingly oscillatory behavior is clearly observable for all *ido* genes but not for *aroH* (Appendix A
Appendix A). Higher peaks of expression are present at 30–40 min and 160–180 min. The *aroH* gene is activated with a different type of kinetic with only picks at 150 min (Appendix A
Appendix A).

Compared to “condition A” (Appendix A
Appendix A), in “condition B” (Appendix A
Appendix A), *aroH* deletion led to an increase of *idoB* and *idoC* expression for both the on and the off experiments. These results show that the two pathways catalyzed by Ido and AroH are probably reciprocally regulated. Indeed, the deletion of *aroH* leads to higher expression of *ido* genes. This was clearer for *idoB* and *idoC* compared to *idoA*. Moreover, the deletion of *idos* increased *aroH* expression. Although this was expected, it is still unclear what the reason for it and what the physiological role of the activation of the two alternative pathways are. Based on the different impacts of metabolites on host immune system, the release of different metabolites (kynurenines or indoles) is extremely of interest.

The model successfully predicted the systems behavior during the *on* experiment: In “condition A”, all genes in the network are inferred as being independently activated in response to Trp (Figure 3A and Appendix A
Appendix A). This suggests that, in the condition with a higher availability of Trp, the different enzymes are independently activated in degrading Trp. In *aro* deficiency (“condition B”), the model detects interactions between *idoA* and *idoC* (Figure 3B). This relationship underlines a determinant action of *idoA* and its derived metabolites in the activation of other genes in the network. This also emerges in the off experiments, where *idoA* is detected as primarily regulating the whole network (Figure 3B).

In addition, the CTBNs-based model also suggests that, in condition of *aro* deficiency, an increased interaction between *ido* genes and increased production of Ido-derived metabolites is possible.

Modeling has become recently a significant tool to understand microbial ecosystem in different contexts, such as clinical microbiology, food fermentation and microbial metabolism. Importantly, modeling may be used to better understand the co-metabolism between the host and pathogens in the context of the immune response [32].

These results also underline the aspect that models are simulated laboratories where it can be artificially improved the experimental design. Future studies, in this particular field, may enhance our understanding on host–pathogen interactions in conditions of nutrient availability. Studies in the field of host–pathogen interactions based on Bayesian networks have been already presented [33]. Those models analyzed the impact of climatic variables or the activation of the immune system during fungal infections in amphibians. Pivotally, in the future, modeling may be used to understand how co-metabolism may affect the outcome of the infection, together with other validated variables.

In particular, in the context of the model presented, the use of additional *Aspergillus* mutants for Trp biosynthesis and catabolism may increase the complexity of the model design.

Based on the result of our study, more comprehensive models based on CTBNs, including, for instance, a detailed construction of Trp metabolism, or based on different experimental conditions, can be developed in the future. This study confirms CTBNs to be an effective methodology for computational gene network reconstruction, and a useful hypothesis generation tool in the study of Trp metabolism: CTBNs could aid in the discovery of novel interactions, supporting the elucidation of those biological processes that strongly impact the fungal fitness, as well as the human immune system during infection.

## Figures and Tables

**Figure 1 jof-06-00108-f001:**
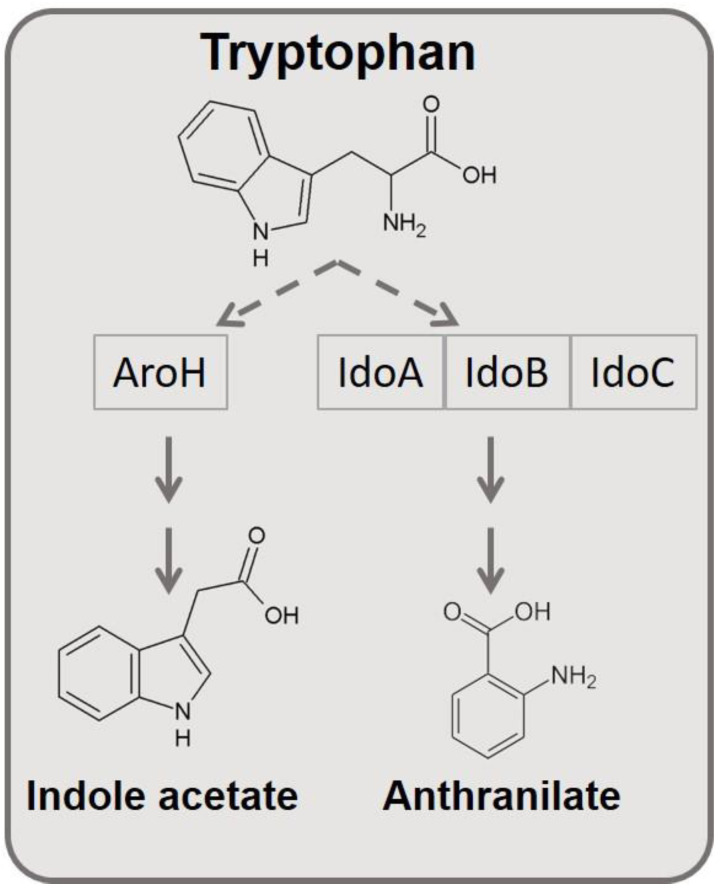
Known literature-based interactions between catabolic axis and the amino acid Trp in *A. fumigatus.*

**Figure 2 jof-06-00108-f002:**
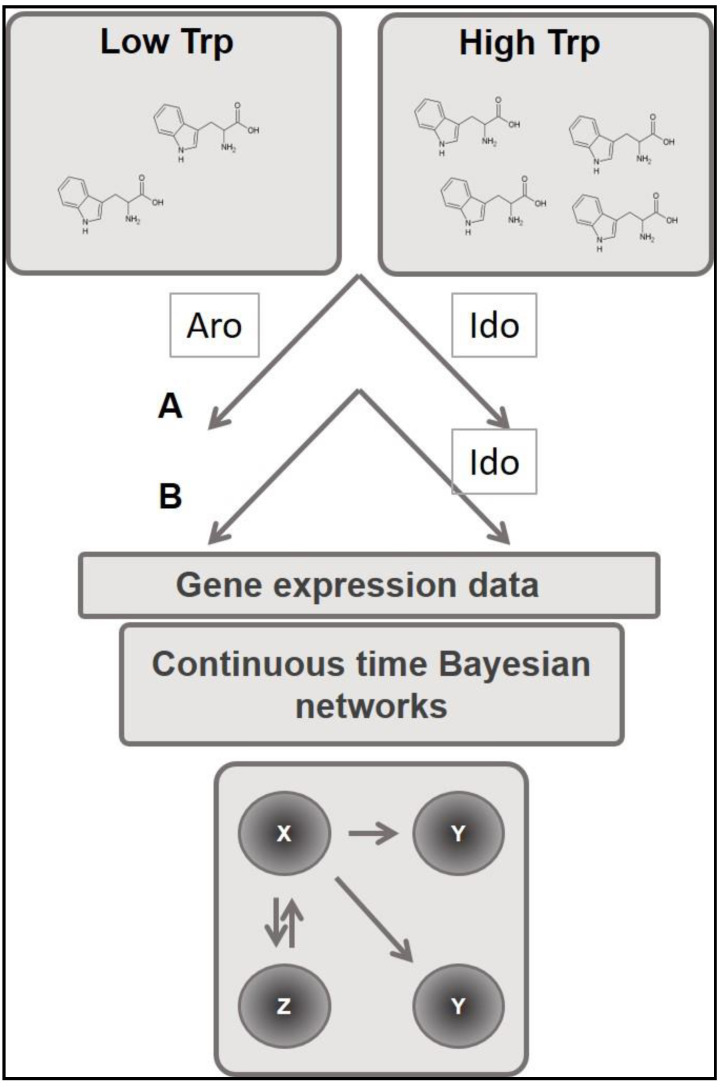
Experimental setting for model construction. A, wild type fungal strain was exposed to different concentrations of Trp (Low or High). B, ∆*aroH* fungal strain exposed to different concentration of Trp (Low or High). X, Y, Z represent generical networks that will be eventually generated in the A and B conditions.

**Figure 3 jof-06-00108-f003:**
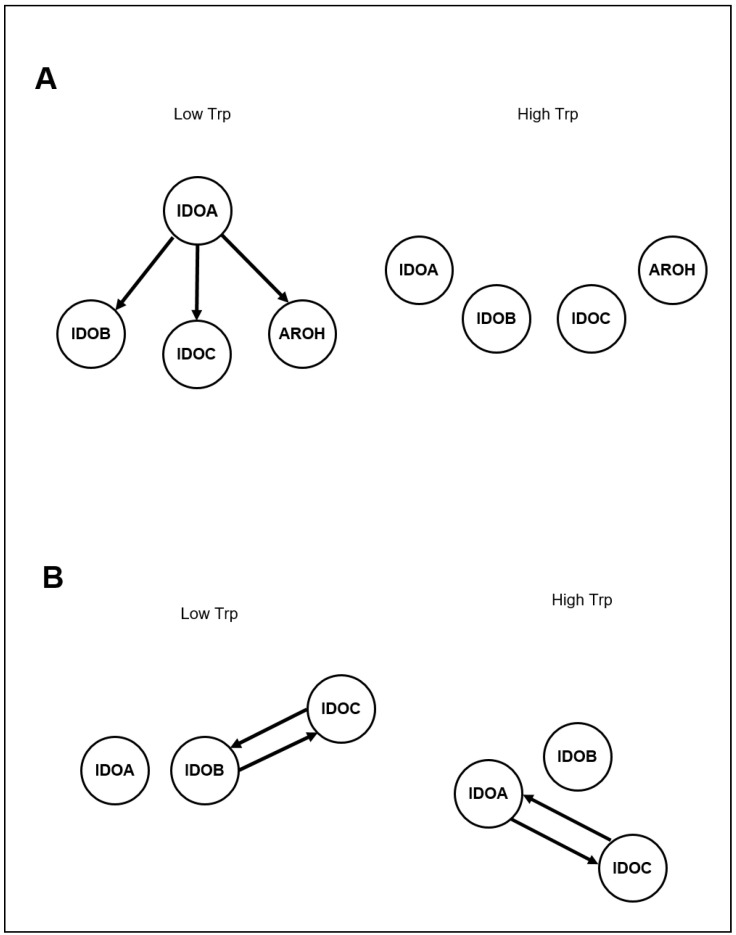
Networks of predicted regulatory interactions learned from RT-PCR time-course data for wild-type strain (Panel **A**), and *aroH* deletion (Panel **B**) experimental conditions. In both cases, separate networks were inferred by using data from low-Trp (off) and high-Trp (on) experiments, for a total of four separate networks.

**Table 1 jof-06-00108-t001:** Fungal Strains used in this study.

ID	Strain	Genotype	Reference
CEA17	CEA17 *pyrG*- KU80	*pyrG1,* ∆*akuB*::*pyrG, pyrG1*	[23]
TTC 22.7	∆*aroH*	*pyrG1,* ∆*akuB::pyrG, pyrG1,* ∆*AFUB_029280*:*pyrG*	This study

**Table 2 jof-06-00108-t002:** qPCR primers used in this study.

Gene	Primers Sequence (5′-3′)	Annealing Temperature (°C)
*18S*	Sense→GAGCCGATAGTCCCCCTAAG αSense→ATGGCCGTTCTTAGTTGGTG	58
*aroH*	Sense→AAAGTCCCGACAGCAATCTACA αSense→TGGGACTTTCACGCTAATCTCT	60
*idoA*	Sense→ATGCCTGTCTCGCTATGC αSense→CTCGGGTGTACGGTTTCG	55
*idoB*	Sense→AGGAAGTTGTCGCTGATTTACC αSense→ATGCTCGCCGCCATTCTG	54
*idoC*	Sense→TCAGCCAGGATGGCAGTC αSense→TCGTCAGTCAGGTCAGGAAG	55

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
