# Peer review of "Modeling Approaches Reveal New Regulatory Networks in Aspergillus fumigatus Metabolism"

_jof, 2020, doi:10.3390/jof6030108_

Round 1

Reviewer 1 Report

Introduction and Merits

In this study, authors have used continuous-time Bayesian networks (CTBN) to infer the dynamic gene interactions during the Trp metabolism in Aspergillus fumigatus. The results from the study could be helpful to manage pathogens during the infections.

Critique

  1. In the introduction section, add the details about other methods that are also comparable to CTBN and why authors decided to use CTBN over other gene network construction methodologies. What is the advantage of CTBN over other highly used gene network tools such as weighted gene correlation network analysis (WGCNA)?
  2. How does CTBN takes into account if the genes are highly expressed or downregulated, and their impact on connecting genes?
  3. In this manuscript, the authors have performed the gene network with very few genes. This will be very unlikely in high-throughput genomics experiments like RNA-seq where we get thousand of significant genes at various time levels. How CTBN is suitable for large-scale data? What is the computational complexity associated with large-scale data? How can you relate this conclusion to small-scale datasets?
  4. Can we deduce the gene interaction strength in CTBN? For example, in gene expression, some genes are highly correlated with some genes than other genes. Can we get this information from CTBN?
  5. In CTBN, what are the assumptions related to gene expression datasets?
  6. In the manuscript, the gene network is inferred using known interaction. How it will predict the interaction with unknown gene interaction? What is the accuracy of network construction using CTBN? How this study can be extended to other metabolic pathways where gene interaction is unknown?
  7. Provide the data used in the analysis. What are the time points for the expression datasets?
  8. Provide the codes used in the analysis in public a repository (e.g. GitHub). It will be useful for readers to reproduce the results obtained in this manuscript.

Reviewer 2 Report

The authors used continuous time Bayesian networks (CTBN) to infer gene regulatory network for tryptophan metabolism in Aspergillus fumigatus.

Time-course gene expression for four genes involved in tryptophan metabolism (aroH, idoA, idoB, and idoC) was measured for wild type and aroH deletion strains in low or high tryptophan conditions (4 genes, 2 strains, 2 conditions), which was used for predicting regulatory interactions.

I think it would help readers if the authors could describe CTBN briefly, define the variables, and explain how the result from structural learning was summarized as shown in Figure 3. I have a few questions and comments for the authors.

  1. Why four separate networks were inferred? Was the condition (low vs high Trp) not considered as a variable in the CTBN? Can the aroH deletion be described as zero expression?
  2. How did the authors define the states for gene expression? Is this 'by discretizing data into 3, 5 and 7 bins of equal size'?
  3. Do the nodes in Figure 3 correspond to the variables in the CTBN? What is the role of 'Low Trp' or 'High Trp' in the CTBN (they are not in circles)?
  4. What is the difference between thick black arrows and thin gray arrows in Figure 3?
  5. How did the authors determine the presence or absence of thick black arrows in Figure 3? Is it from structural learning? 
  6. 'Only arcs detected in all candidate networks that had at least one arc were considered to be high-confidence and reported in the final graphs shown in Figure 2.' Figure 3 instead of Figure 2. How many candidate networks had no arc? Was there no arc at all in Figure 3A 'High Trp'?
  7. How did the authors determine the presence or absence of the thin gray arrows in Figure 3?
  8. Can the authors provide the data, R codes, and results for reproducibility?
